# Data-Driven Sparse Sensor Placement Optimization on Wings for Flight-By-Feel: Bioinspired Approach and Application

**DOI:** 10.3390/biomimetics9100631

**Published:** 2024-10-17

**Authors:** Alex C. Hollenbeck, Atticus J. Beachy, Ramana V. Grandhi, Alexander M. Pankonien

**Affiliations:** 1Air Force Institute of Technology, Dayton, OH 45433-7765, USA; 2Air Force Research Laboratory, Dayton, OH 45433-7765, USA

**Keywords:** optimization, sparse sensing, flow sensors, flight control, artificial hair sensors

## Abstract

Flight-by-feel (FBF) is an approach to flight control that uses dispersed sensors on the wings of aircraft to detect flight state. While biological FBF systems, such as the wings of insects, often contain hundreds of strain and flow sensors, artificial systems are highly constrained by size, weight, and power (SWaP) considerations, especially for small aircraft. An optimization approach is needed to determine how many sensors are required and where they should be placed on the wing. Airflow fields can be highly nonlinear, and many local minima exist for sensor placement, meaning conventional optimization techniques are unreliable for this application. The Sparse Sensor Placement Optimization for Prediction (SSPOP) algorithm extracts information from a dense array of flow data using singular value decomposition and linear discriminant analysis, thereby identifying the most information-rich sparse subset of sensor locations. In this research, the SSPOP algorithm is evaluated for the placement of artificial hair sensors on a 3D delta wing model with a 45° sweep angle and a blunt leading edge. The sensor placement solution, or design point (DP), is shown to rank within the top one percent of all possible solutions by root mean square error in angle of attack prediction. This research is the first to evaluate SSPOP on a 3D model and the first to include variable length hairs for variable velocity sensitivity. A comparison of SSPOP against conventional greedy search and gradient-based optimization shows that SSPOP DP ranks nearest to optimal in over 90 percent of models and is far more robust to model variation. The successful application of SSPOP in complex 3D flows paves the way for experimental sensor placement optimization for artificial hair-cell airflow sensors and is a major step toward biomimetic flight-by-feel.

## 1. Introduction

Aircraft design, perhaps more so than any other modern science, has its roots firmly planted in bioinspiration. However, until very recently, practical and technological constraints have limited the extent of biomimicry. Birds and bats have continuously morphing wings with seemingly infinite configurations [1,2,3,4,5]; aircraft have slots, flaps, and, more rarely, active sweep. Bats and insects have hundreds or thousands of sensory hairs through which they feel the airflow; aircraft have one to several pitot probes and static ports. Recent advancements in materials and actuators have made continuously variable camber, twist, and planform possible [6], which may bring technology closer to matching the performance of natural flyers [7]. These developments in morphing wing and sensor technology, as well as concurrent advances in understanding of extant [8] and extinct [9] animal flight, may be ushering in a new era of dexterous, nimble aircraft whose control systems are lighter, more robust, faster, and more information-rich than those of conventional aircraft [10]. A bioinspired systems-based approach is needed [11]; conventional sensors and control systems may be unreliable or infeasible for these emerging aircraft shapes, actuators, and integrated control surfaces [12,13].

A bioinspired Flight-By-Feel (FBF) system may employ integrated arrays of any number of pressure, strain, and flow sensors to enable rapid and agile flight state estimation and response [14]. Flying animals provide the proof-of-concept that FBF control is feasible, although the intricacies of their flight control systems remain poorly understood [15]. The challenge for aircraft designers, constrained by size, weight, and power (SWaP) budgets, is to determine where best to place a limited number of sensors to detect the most useful information from the airflow over a wing or aircraft body. Optimal sensor placement may allow a handful of sensors to capture as much information as a dense array of randomly placed sensors, doing the same job but with less lag in closed-loop control, lower SWaP central processors, and easier integration. Sensor placement optimization can also be coupled with actuator design or placement optimization to achieve a lightweight structure capable of stability and control [16].

The functions describing the airflow over a wing in dynamic conditions are highly complex (high-dimensional), nonlinear, and discrete, which pose severe difficulties for conventional optimization approaches. Gradient-based optimization of discrete data requires pre-processing of the data to create a pseudo-continuous approximation, and in the case of aircraft wings must account for many discontinuities—edges, corners, intersections, and other constraints. A greedy search is simpler to implement in complex nonlinear models but is even more susceptible to local minima. These techniques may achieve a near-optimal solution, but there is no robust method to ensure such a solution for complex models where the true optimum cannot be determined by calculation or by force. Problems of this type are considered NP-hard; the search space is vast, the functions are unknown or unknowable, and there is no deterministic solution [17,18]. A brute-force search of the design space, while feasible for small models, as demonstrated here, is prohibitive or impossible for realistic sensor placement problems. The present work demonstrates that the data-driven Sparse Sensor Placement Optimization for Prediction (SSPOP) algorithm, recently introduced for 2D airfoils [19,20,21], is capable of reliably finding a top-one-percent Design Point (DP), or sparse set of sensor locations, for any number of sensors on a 3D wing, as ranked by accuracy in predicting the angle of attack (AoA or α) from airflow velocity magnitude data. The performance of the SSPOP DP is compared against the best possible DP where a combanitorial brute force search is possible, and against conventional optimization approaches, including a greedy search and a gradient-based auto-differentiated technique.

The SSPOP algorithm used here is an adaptation of the Sparse Sensor Placement Optimization for Classification (SSPOC) algorithm pioneered by Brunton et al. [22] and later expanded for Reconstruction (SSPOR) by the same group [23,24,25]. These optimization algorithms all use data reduction techniques to greatly reduce the dimensionality of high-order systems. Like artificial hair sensors, data reduction for sensing is bioinspired. Animals interact with high-dimensional physical systems via limited sensory information, a form of compressive sensing of big data [26,27], and process diverse stimuli with limited sensor types as multi-modal signals [28,29].

In the present research, the sensors in the modeled FBF system were based on bat-like artificial hair-cell flow sensors (AHS), which detect velocity magnitude from mechanical drag force on hair-like structures (See [30] for bat hair-cell morphology and function, and [31,32] for a bat-like AHS). Some models included variable-length hairs, as hair length has been shown to have a significant impact on sensitivity [33,34]. A recent survey by the authors describes and evaluates hair-type and similar flow sensors to synthesize sensor design, function, placement optimization, FBF control, and next-generation aircraft design into a cohesive bioinspired research paradigm [19].

Most of the work in distributed sensing for FBF has involved the tractable cases of 2D airfoils and rectangular wings on conventional-style aircraft, and rarely consider optimal placement (for a recent example of a square array of AHS used for flap-morphing gust alleviation, see [35]). This leaves the realms of complex flows, unsteady aerodynamics, and vortical flow largely unexplored. The flow over highly-swept wings, such as in the delta wing model used in the present research, includes prominent leading edge vortex (LEV) features, a common flow-control and lift-generating phenomenon found in nature [36,37,38,39,40,41,42] and increasingly in aircraft [43]. Figure 1 compares physical and computational visualizations of a flapping dragonfly and a delta wing fighter aircraft, showing that both exhibit significant vortical flow structures. Flight control for aircraft with persistent LEVs is highly challenging [44], making a delta wing a prime candidate for evaluating the effectiveness of sensor placement optimization for FBF in complex flows.

This work represents the first application of a data-driven approach for optimal placement of flow sensors on 3D wings, and the first application using variable-length hairs. These results confirm the conclusion from previous 2D studies ([19,20]) that the SSPOP algorithm is flexible in scale and scope, with promising FBF implications for sensors of any type and aircraft of any size. The successful application of SSPOP to 3D wings is a step toward solving the “grand challenge problem” of generalized optimization with scalability, which may enable rapid advancements in a broad range of engineering and system designs [23,24,25].

## 2. Materials and Methods

This paper covers the application and evaluation of the SSPOP algorithm for flow sensor placement on a delta wing. The following subsections describe the wing model, the SSPOP algorithm, and the brute-force search approach for verification of the results.

### 2.1. Wing Model

The model was an NACA 4415 delta wing with 45° sweep and a root chord of 250 mm. The wing geometry was constructed in Ansys DesignModeler as a half-span wing in a tunnel setting. Ansys Workbench and Ansys Fluent were used to perform computational fluid dynamics (CFD) modeling to acquire velocity data over a wide range of α. Ansys Fluent was used for grid generation of a polyhedral volume mesh limited to approximately 500,000 cells for efficiency. Turbulent steady-state estimations of the flow field at 10 m/s freestream velocity were solved for each α from −10° to 20° at half-degree increments using constant density gas at standard conditions with a two-equation k-ω Shear-Stress Transport model. The velocity was selected to match the nominal sensitivity of real artificial hair sensors to be used in future experiments. Each case was solved using a pseudo-transient pressure-based solver where the momentum equation and pressure-based continuity equations were solved simultaneously. This procedure was validated by modeling a similar wing matching the geometry and conditions of an experimental wind tunnel test from [47], finding errors of approximately 2% for the coefficients of lift ($c_{l}$) and drag ($c_{d}$) and 0.20% for lift-to-drag ratio [20].

Velocity magnitude (not direction, as many AHS types are isotropic [19] and many natural flow-sensing systems measure only velocity [48]) was calculated at 0.5 cm, 1.0 cm, and 2.0 cm above the surface along 11 chordwise slices of the wing (Figure 2). This resulted in a three-layer shell of velocity data from which data sets were extracted with variable AHS height at 238 nodes (candidate sensor locations) over the top and bottom surfaces of the wing (Figure 3). This allowed SSPOP analysis for three fixed-height cases as well as a variable height case, which corresponded to real-world AHSs in which the length of the hair may be tuned during manufacturing or installation to account for different sensitivities [31,49]. Nodes were spaced approximately 1 cm apart.

Nodes can be considered binary variables. For any given DP with *Q* sensors and *N* nodes, there will be *Q* local measurements and N−Q nodes with null measurements; the DP surface velocity distribution is sparse, while the full data set is semi-continuous. When linear regression is performed on the variation of velocity at each DP node over a range of conditions, the prediction model will have *Q* nonzero coefficients and N−Q coefficients of zero value. The solution space poses a combinatorial problem: for any given *Q* and *N* there are $\frac{N!}{Q!(N-Q)!}$ possible DPs. For small models with few sensors and relatively few nodes, a brute force search of the design space can be performed to find the optimal DP. However, realistic models may have many billions of DPs, so a data-driven approach to find a near-optimal DP is necessary.

### 2.2. Finding and Testing the Design Point

The SSPOP algorithm bypasses this combinatorial problem by greatly reducing its dimensionality. SSPOP (Figure 4) uses singular value decomposition (SVD) and linear discriminant analysis (LDA) to identify a sparse set of sensor locations (a design point (DP)) at which most of the information from the flow may be obtained. It begins with SVD of the near-surface velocity magnitude data to identify prominent flow field features and create a low-rank approximation of the original data set. The data are then truncated using a weights matrix via LDA based on the variances of the original data. The highly truncated data sets contain well over 95% of the information of the full dataset. A convex solver then creates the final sensor matrix and identifies the corresponding DP. Finally, linear regression of velocity data at the DP sensor locations is performed to obtain a root mean square error (RMSE) of prediction for any flight parameter (for example, angle of attack, α). It should be emphasized that SSPOP DP is application agnostic; the algorithm selects the most information-rich DP regardless of what that information will be used to predict. The RMSE between the original data (CFD simulation) and the sparsely calculated linear regression prediction is the performance metric of the DP. To determine its proximity to optimality, the performance of the SSPOP DP in terms of predictive accuracy was evaluated through comparison with all possible DPs found by a brute-force search. Analysis by this method is limited to low *Q*, especially when *N* is large. Once validated in general, the predictive performance of any SSPOP DP can be used to baseline a search of all possible DPs for a given model. See Section 5 for a link to supplementary information including the Matlab code and detailed flow chart for the SSPOP algorithm.

## 3. Results

The SSPOP algorithm found a DP for one through four sensors for each of the three single-length AHS models, and for one through three sensors for the variable-length model. A simple search was performed to improve the SSPOP performance, and all cases were compared against the best possible DP found by brute-force search. The results are presented in Table 1 and described below, followed by a comparison of SSPOP against conventional optimization methods for one through ten sensors for each model.

### 3.1. Design Points and Predictive Performance

The predictive performance by RMSE of α and relative ranking of the SSPOP, SSPOP + Search, and Best Possible (BP) DPs are given in Figure 5. Among the single-length cases, the 1 cm AHS case performed best in all metrics for the three- and four-sensor DPs. This may have been anticipated, as the airflow at this height may contain a wider range of information from both within and outside the boundary layer, compared with the shorter or longer hairs, which may be too far under the viscous boundary or too far into the freestream, respectively. Also, as expected, the variable length model had the best brute-force optimum performance for all *Q*, due to having a design space three times as large. Unexpectedly, however, the 1 cm model outperformed the variable model in SSPOP and SSPOP + Search for the three-sensor case, possibly due to the greater complexity of the variable dataset. In the brute-force search three-sensor case, the variable model retained one of the 1 cm hair nodes but replaced the other two with 2 cm hairs in nearly the same location but on the opposite surface. These changes in the variable model did not result in a significant improvement in performance (0.043° versus 0.039° RMSE α) over the 1 cm model. This result indicates that the performance gains, if any, from variable length AHS models might not be worth the cost of manufacturing hairs of various lengths. Identifying a single ideal hair length for a given airfoil or wing may be advantageous. The SSPOP algorithm can assist in optimizing hair length through rapid iterations of different hair length models.

The SSPOP + Search results suggest diminishing returns with this method as *Q* increases. As the size of the design space increases, the rate of gain beyond the SSPOP performance from searching the design space slows considerably. This indicates that the SSPOP + Search method might not add value to SSPOP alone in large models. For instance, the three-sensor SSPOP + Search case did worse than the two-sensor case because the search was more difficult in higher dimensions. While the three-sensor case started with a better SSPOP solution, the search did not improve the solution enough and it fell behind.

Figure 6 depicts the sensor locations for the SSPOP and BP DPs. Each row represents a chordwise slice, with Row 1 nearest the wing root and Row 11 nearest the wing tip. The thickness of the NACA 4415 airfoil was exaggerated by a factor of four to facilitate visualization. The BP DP was found for up to four sensors for each fixed hair length case. The variable hair length case was only characterized for three sensors because the design space (238 nodes × 3 lengths = 714 variables and 10,738,066,626 DPs for a four-sensor solution) was too large for a brute-force search. The SSPOP DPs for all single-length models (a–c) included multiple sensors near the leading edge, while the BP sensor locations were, on average, closer to the trailing edge. The single-length BP DPs tended to include sensors in more extreme locations, nearer the wingtip and trailing edge. However, the SSPOP DPs were more likely to include neighboring sensors. The variable length model (d) featured the closest agreement between the SSPOP DP and the BP DP. Two-thirds of the SSPOP sensors were located on the top, compared with only one-third of the BP sensors.

### 3.2. SSPOP Versus Conventional Optimization

The simplicity and efficiency of the SSPOP algorithm is a major driver in its use for sensor placement optimization over conventional approaches. This section compares SSPOP DP performance against greedy search (GS) and then against gradient-based optimization using auto-differentiation, evaluating the costs and benefits of each method.

#### 3.2.1. Greedy Search

A GS optimization algorithm first finds the best single-sensor solution, then finds the best two-sensor solution in which the location of the first sensor is retained, and so on. The benefits of a GS is its simplicity and speed, and there is no upper limit for any realistic model in terms of the node and sensor quantity. For a model with linear regression such as that used here, each iteration of the search has negligible cost, and the total cost of the GS is also negligible. The question is whether GS reliably arrives at a near-optimal solution for any *Q*. It cannot be known with certainty whether a GS solution approaches the optimum unless we compare it to a brute force search, but this is infeasible for models with high *Q* and high *N*. However, having shown that the SSPOP solutions are within the top 1% of all possible solutions where the design space can be characterized, it is valid to compare the relative performance of SSPOP and GS solutions for any number of sensors.

Figure 7 plots the predictive performances of the SSPOP and GS DPs for one through ten sensors for an airfoil model and 2D slice of a wing (model from [20]), and the 3D wing of the present study. In most cases, the SSPOP solution outperforms the GS solution, especially at lower *Q*. In other instances at low *Q*, and in all cases for high *Q*, the GS solution approximately matches the SSPOP solution. In no case does the GS solution significantly outperform the SSPOP solution.

Based on these results, it is evident that a GS *can* yield a top-one-percent result, especially at higher *Q* values, but some form of validation would be required to confirm this for each model. We have shown in [20] and above that SSPOP reliably finds a top-one-percent solution, regardless of model, in both 2D and 3D for two or more sensors. The two approaches may be used together, as done earlier with SSPOP + Search. The SSPOP + GS would first run SSPOP to establish a goal, and either a GS or simple search, or both, could be run to seek improvement. From Figure 8, it is evident that the SSPOP + Search approach delivers diminished returns as *Q* increases, due to the exponentially increased design space at larger *Q*. Therefore, for very large models, it may be sufficient to simply keep the SSPOP DP without additional searches.

The data-driven SSPOP algorithm may be less sensitive to model configuration than GS. Figure 8 shows the ten-sensor GS and SSPOP DPs for two variations of an airfoil model. Compared with SSPOP, GS more significantly exploits the extreme conditions at the leading and trailing edges, especially for the full model. Tight groupings that may be impractical for real-world sensor installation are more prevalent in GS DPs. In addition, GS is sensitive to local minima in the objective function, which may lead to early "mistakes" in node selection, which impact the higher-*Q* models. For example, the best single-sensor solution is highly variable in performance and tends to be located at extreme points. In a GS, this sensor must be retained, whether it is helpful in combination with the others or not.

#### 3.2.2. Other Conventional Optimization Approaches

The SSPOP and brute-force methods formulate the sensor placements as a discrete problem, where the sensors must be placed at specific node locations. However, it can also be formulated as a continuous problem by interpolating the velocities between the grid points. Given specific sensor locations, one can perform linear regression on the training cases and evaluate accuracy on the test cases. This entire pipeline of sensor locations → sensor velocities → linear regression to predict α from sensor velocities → predicting α in the test cases and finding the MSE was implemented in TensorFlow (Figure 9). This allows the gradients of α-prediction-error with respect to sensor locations to be found using auto-differentiation, enabling the application of gradient-based optimization algorithms.

Linear regression is performed using TensorFlow’s tf.linalg.lstsq function, which requires fast=True in order to be differentiable. However, this setting causes the linear regression to use Cholesky decomposition. While this method is less robust and stable than the complete orthogonal decomposition used when fast=False, an L2 regularization of $10^{-7}$ aided numerical stability.

Sensor locations were constrained to always remain within the appropriate region of the airfoil surface. Whenever a sensor moved outside of the feasible region, it was both penalized and moved to within the sensor region before linear regression was performed, ensuring the optimizer settled on a feasible solution.

The continuous optimization formulation introduces a limitation not found in the discrete optimization formulation of SSPOP. Because sensors cannot be added too close to the leading or trailing wing edges, the feasible regions on the top and bottom of the wing are separated. This means sensors will remain confined within their initial regions. To handle this, we ran two different optimizations for each hair length case, one with all three discrete variable length sensors on the top of the wing and one with all three discrete variable length sensors on the bottom.

Two optimization algorithms for sensor placement were initially considered: Adam (adaptive moment estimation) and BFGS (Broyden–Fletcher–Goldfarb–Shanno). It was found that Adam is less reliable than BFGS for this use case. The Adam solution tends to fluctuate unpredictably, and can move away from a good solution without returning. This behavior is useful when training a neural network with thousands-to-billions of parameters, where the parameter space is full of nearly-equally-good local optima and the most important thing is to avoid getting stuck at one of the many saddle points. However, in engineering applications, BFGS performs better. Therefore, only BFGS results are compared to SSPOP. For implementation, we used SciPy’s L-BFGS-B function, where the L stands for limited memory and the B stands for box constraints (where each variable is constrained by a minimum and maximum value). BFGS approximates the Hessian of the loss function using gradients calculated with auto-differentiation.

In total, we ran four cases, with 5 mm, 1 cm, 2 cm, and variable length hairs. For this last case, the hair lengths were optimized as continuous variables. Because BFGS is not a global optimization method, it was run multiple times. For each optimization case, 10 different initial sensor placements were selected using Latin Hypercube Sampling, a BFGS optimization run was conducted for each initial sensor placement, and the best optimum from the 10 results was selected as the final optimum. Finally, the sensors at this final continuous optimization are snapped to the nearest grid points. In cases where one or more sensors are nearly equidistant from multiple grid points, all combinations of nearby grid points are tested and the best configuration selected. All results are shown in Table 2. BFGS outperforms SSPOP in the 5 mm case, but performs worse in the other three cases.

Snapping the sensors to the grid resulted in varying levels of degradation for the α prediction accuracy. For instance, in the case with 5 mm hairs on top of the wing, the test error increased negligibly from $0.143^{\circ }$ to $0.145^{\circ }$. At the other extreme, adjusting the 1 cm sensors on the top of the wing increased error from $0.137^{\circ }$ to $0.514^{\circ }$, or a factor of four.

On average, each of the 10 optimization runs for each case took 15–20 s to complete, making the method fast to execute. This makes BFGS comparable in speed to SSPOP, even if the optimization is repeated 10 times to increase the chance of finding the global optimum.

On the other hand, BFGS with auto-differentiation also comes with drawbacks in this application. First, it is more difficult and time-consuming to implement than SSPOP, requiring interpolation of air velocity values, geometric constraints on sensor locations, a differentiable implementation of linear regression, and the testing of multiple combinations of nearby grid points once optimization converges. While none of these individually are too difficult to implement, they do take time, which is not required when implementing SSPOP. Second, each sensor is constrained to its initial region. While this can be handled by running optimization cases for every possible combination of sensors in every region, this scales poorly as the number of sensors and number of regions increases. Finally, as previously mentioned, the accuracy of the final BFGS solution is often highly sensitive to small perturbations in the sensor location and may be sensitive to the initial sensor location choice, unlike SSPOP. Despite these drawbacks, BFGS with auto-differentiation performed only slightly worse than SSPOP in this example, and either method is powerful enough to handle sensor placement optimization.

## 4. Discussion

The SSPOP algorithm is capable of quickly finding a DP of AHS placements that predicts α by linear regression to well within 0.10° accuracy over a wide range of α for 3D wings. The performance of these SSPOP DPs generally ranks within the top 1% of all possible DPs for those cases where a brute force analysis was performed, granting confidence that similar performance can be expected for any arbitrary number of sensors and nodes on a wing of any shape and size. Therefore, in large models for which a brute-force search is infeasible and the true optimum DP cannot be found, we can use the SSPOP DP as a top-one-percent solution starting point. If needed, a search can then be performed to find a DP which performs up to an order of magnitude better. The SSPOP solution matched or outperformed conventional greedy and gradient-based optimization techniques, especially for very low numbers of sensors. These results have significant implications for bioinspired aircraft design, sensor design and placement, and FBF control system design.

### 4.1. SSPOP and Flight-By-Feel

In general, increasing the number of sensors *Q* improves the achievable predictive accuracy. However, the predictive performance varies significantly across the possible DPs for a given *Q* and *N*. In order to find a DP suitable for FBF, an objective maximum RMSE (for AoA or any flight parameter) may be set based on the requirements of the FBF controller and desired performance. In the present work, no RMSE performance objective was given, other than to find the best possible performance subject to the constraints of the solution method. We have shown that the SSPOP algorithm can quickly predict the near-optimal performance of any number of sensors on a wing, so a flight control designer could incorporate these metrics into a tradeoff analysis for the number of sensors. After performing SSPOP for a range of *Q* (as in Figure 7), the DP with the lowest *Q* meeting a control-system-driven predictive accuracy requirement would be considered to be the optimum for the given criteria. Comprehensive FBF systems for future autonomous aircraft might benefit from a combination of sensor placement optimization approaches for different sensor types and modalities. For example, a flapping-wing MAV might use strain sensors placed by an observability analysis optimization approach on the wings for wing state estimation [50], SSPOP could optimally place hair sensors on the body for flight state estimation, bioinspired isolators could protect sensitive navigation equipment [51], and nano-scale hair sensors could be used for inertial and directional control [52].

### 4.2. SSPOP and Aircraft Design

SSPOP only requires a discrete dataset of flow, pressure, or other metrics, which can be found by computer modeling (such as CFD) or wind tunnel testing (such as particle image velocimetry). Therefore, SSPOP can be used early and often in the design process, helping to determine the required number and placement of sensors, even as early as the conceptual design stage. The flexibility and robustness of the SSPOP algorithm is potentially a major advantage over conventional sensor placement optimization approaches, especially early in the design process when configurations are subject to significant change. Unlike conventional approaches in which sensors are placed only after the aircraft design is fixed, a bioinspired FBF design approach can use SSPOP to ensure the appropriate sensors are seamlessly integrated into the skin of the aircraft in a near-optimal location. Likewise, the form and structure of an AHS itself may be varied early in the design process, as we have done with our variable hair length model. Just as we observed in the hair sensors of natural fliers, an AHS should be fine-tuned to suit the perception needs for flight control based on its location and envelope of flight conditions [49,53].

### 4.3. Future Work

The SSPOP algorithm should be applied to more complex FBF problems. The prediction of flight state should be expanded beyond AoA to a wider range of relevant metrics, such as sideslip angle, roll angle, rates of rotation on all axes, and aerodynamic coefficients. Additionally, as the SSPOP DP is based on data alone and is application-agnostic, nonlinear regression or artificial neural networks might be considered for prediction.

SSPOP should also be applied to other types of sensors and applications, such as structural, thermal, or optical systems. For example, it may be used to place sensors to optimally capture the stress and strain or thermal load on a critical structural component to prevent system failure. An optical SSPOP solution might optimally predict the future behavior of a system from sparsely sampled images of its physical state over time. Any system that can be modeled or measured in high dimensions can be reduced by SSPOP to a sparse approximation containing nearly all of the original information, and this sparse set of measurements may be used to predict a wide array of useful metrics about the system.

Real-world applications for SSPOP include novel configuration/control designs for aircraft or aquatic vehicles, development of novel sensors, and distribution configurations supporting bioinspired autonomous control-by-feel. By operating on raw data alone, and not being sensitive to initial or boundary conditions, SSPOP opens a new path for optimization in areas where it had previously been prohibitively costly or impossible. The SSPOP approach can quickly predict the performance of any number of sensors, so a flight control design process could incorporate this algorithm in a tradeoff analysis. After performing SSPOP for a range of *Q*, the DP with the lowest *Q* meeting a control system predictive accuracy requirement would be selected.

## 5. Conclusions

This research tested a data-driven approach for identifying near-optimal locations of a sparse set of sensors. In the example computational model, artificial hair sensors captured air velocity magnitudes in order to determine the angle of attack of an NACA 4415 45°-swept delta-wing. The performance of the DP selected by the SSPOP algorithm was compared, where possible, to the true optimum DP found by brute-force search. For two or more sensors, the SSPOP DP ranked within or near the top 1% of all possible DPs by RMSE in predicting AoA, and achieved accuracy near 0.10° error for four or more sensors. A simple search procedure performed after SSPOP yielded a DP well within the top 0.01% in a few minutes versus the several hours required for a brute-force search for these relatively small models. The SSPOP algorithm is highly adaptable, having been previously demonstrated on 2D models and, here, on three fixed-length AHS models and one variable-length AHS model—performing well in all cases. Therefore, SSPOP can be applied in two or three dimensions for any shape and node configuration and for pressure, strain, or any other FBF-relevant sensors. SSPOP is well suited for NP-hard problems such as sensor placement on aircraft. Unlike conventional optimization techniques, SSPOP does not depend on node/region continuity, allows for discrete flow data, and is not sensitive to initial conditions. By quickly and easily guiding the placement of sensors for near-optimum predictive accuracy, the SSPOP algorithm can support the implementation of FBF control in the design and operation of next-generation aircraft. The success of SSPOP for bioinspired FBF control suggests the applicability of bioinspiration in aircraft design extends far beyond its current horizons.

## Figures and Tables

**Figure 1 biomimetics-09-00631-f001:**
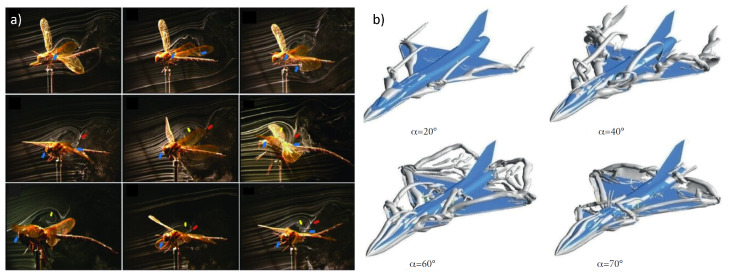
Lift-inducing LEV formation and evolution in (**a**) flapping dragonfly wings (from [45]) and (**b**) a delta-canard fighter at high AoA (from [46]).

**Figure 2 biomimetics-09-00631-f002:**
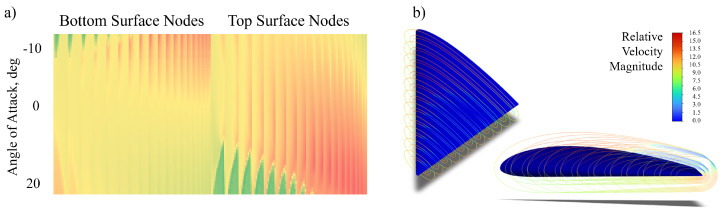
45 deg sweep, NACA 4415 delta wing model. (**a**) Matrix of airflow velocity magnitude for the 1cm hair length model at each of 238 nodes (columns) over 40 angles of attack (rows) from −10° to 20°. (**b**) CFD°: Velocity magnitude at three heights above wing for 11 slices at $\alpha =20^{\circ }$.

**Figure 3 biomimetics-09-00631-f003:**
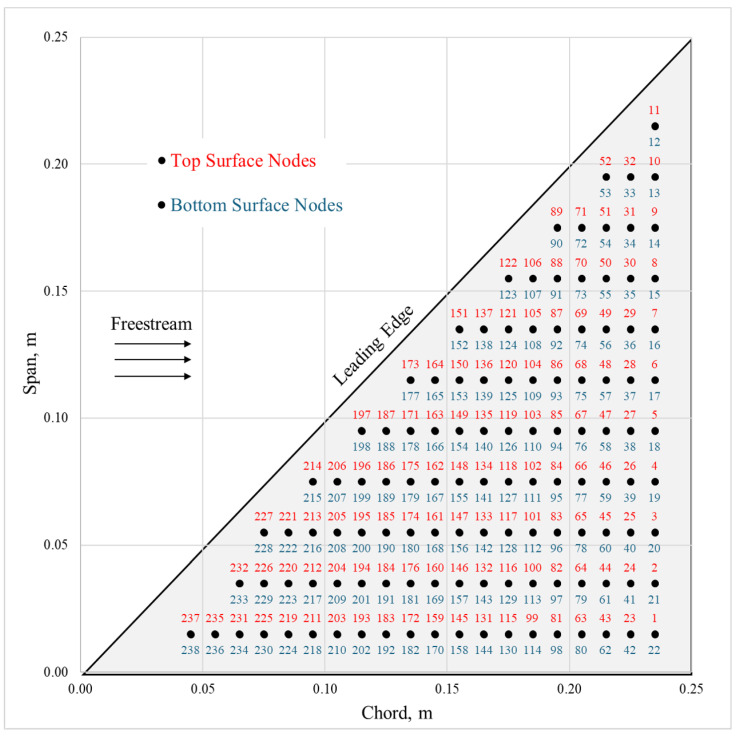
Node locations and numbers. There are 238 nodes spaced evenly over the top and bottom surfaces.

**Figure 4 biomimetics-09-00631-f004:**
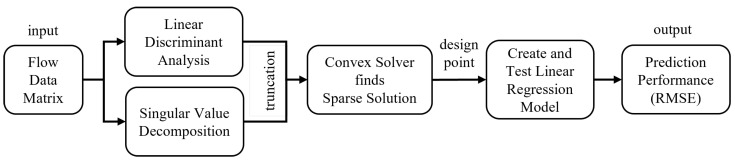
The Sparse Sensor Placement Optimization for Prediction (SSPOP) algorithm. From [20].

**Figure 5 biomimetics-09-00631-f005:**
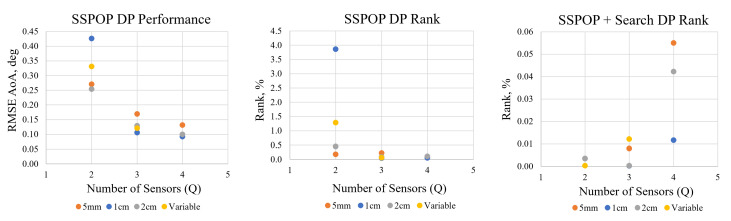
Performance and ranking for the three individual hair length models and the variable hair length model.

**Figure 6 biomimetics-09-00631-f006:**
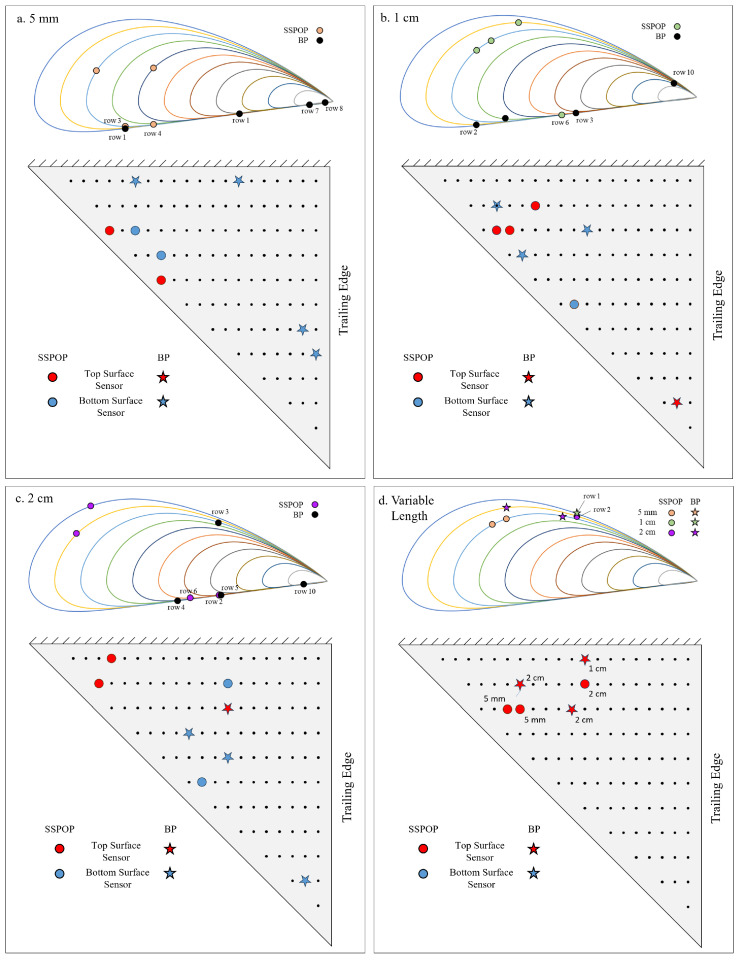
Design points for SSPOP and best possible with four sensors for (**a**) 5 mm, (**b**) 1 cm, (**c**) 2 cm, and (**d**) three sensors for variable AHS hair lengths.

**Figure 7 biomimetics-09-00631-f007:**
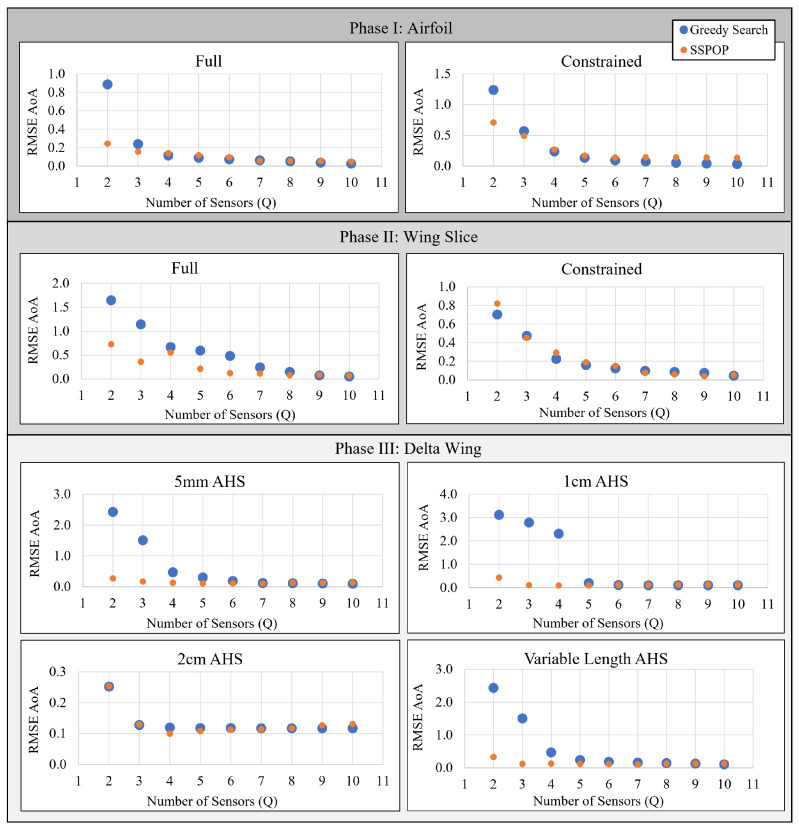
DP performance (degrees AoA prediction) of a greedy Search and SSPOP for two through ten sensors.

**Figure 8 biomimetics-09-00631-f008:**
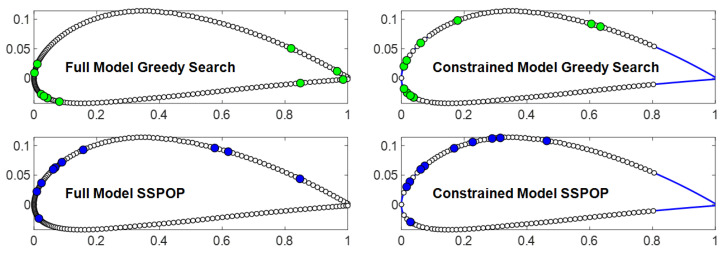
Ten-sensor design points for SSPOP and greedy search solutions for a 2D airfoil (model from [20]).

**Figure 9 biomimetics-09-00631-f009:**
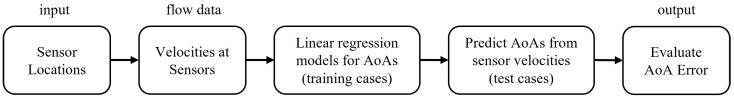
Flowchart of auto-differentiated sequential data processing.

**Table 1 biomimetics-09-00631-t001:** Design points, performance, and ranking for three single-hair-length models and one model with variable hair length. Nodes for the variable model are colored according to their length.

		SSPOP						SSPOP + Search						Best Possible				
	Q	DP				RMSE (deg)	Rank (%)	DP				RMSE (deg)	Rank (%)	DP				RMSE (deg)
5 cm	1	227				7.517	2.521							13				7.105
	2	216	227			0.271	0.177	183	211			0.142	0.004	183	211			0.142
	3	43	227	228		0.170	0.231	188	193	225		0.137	0.008	130	198	207		0.095
	4	197	199	216	227	0.131	0.105	130	142	158	232	0.125	0.055	15	36	130	218	0.045
1 cm	1	227				7.756	1.261							227				7.493
	2	215	227			0.426	3.865	21	224			0.125	0.004	210	224			0.125
	3	153	213	232		0.106	0.048	156	161	223		0.060	0.000	145	167	208		0.043
	4	165	194	213	221	0.092	0.057	86	93	210	223	0.075	0.012	32	156	207	223	0.036
2 cm	1	227				8.009	0.420							227				8.009
	2	133	199			0.254	0.450	145	230			0.161	0.004	145	230			0.161
	3	141	180	214		0.129	0.091	155	200	220		0.064	0.000	33	96	188		0.056
	4	143	165	225	232	0.100	0.111	112	157	189	218	0.088	0.042	33	133	140	178	0.038
Var.	1	227				7.517	0.980							13				7.105
	2	64	215			0.331	1.288	156	223			0.080	0.000	156	223			0.080
	3	146	205	213		0.121	0.061	206	210	234		0.097	0.012	145	161	204		0.039

**Table 2 biomimetics-09-00631-t002:** Comparison of sensor locations and performance for optimum (brute-force search), full-surface SSPOP, and top- or bottom-only BFGS DPs. Rows are ordered from best-to-worst RMSE.

	Method	RMSE deg α	DP Nodes					Method	RMSE deg α	DP Nodes			
5 mm	BF Search	0.045	15	36	130	218	2 cm	BF Search	0.038	33	133	140	178
	BFGS (bot)	0.099	22	109	168	210		SSPOP	0.100	143	165	225	232
	SSPOP	0.131	197	199	216	227		BFGS (top)	0.143	1	7	23	226
	BFGS (top)	0.145	1	9	137	235		BFGS (bot)	0.150	57	93	125	217
1 cm	BF Search	0.036	32	156	207	223	Var.	BF Search	0.039	145	161	204	
	SSPOP	0.092	165	194	213	221		SSPOP	0.121	146	205	213	
	BFGS (bot)	0.141	40	41	140	190		BFGS (bot)	0.123	19	98	207	
	BFGS (top)	0.514	1	23	115	175		BFGS (top)	0.966	1	23	131	

## Data Availability

A Matlab code supplement is available [54] for reproducing some of the results in this manuscript, including the SSPOP algorithm code, example flow datasets, and detailed flow charts of the SSPOP algorithm.

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
