# Peer review of "Data-Driven Sparse Sensor Placement Optimization on Wings for Flight-By-Feel: Bioinspired Approach and Application"

_biomimetics, 2024, doi:10.3390/biomimetics9100631_

Round 1

Reviewer 1 Report

Comments and Suggestions for Authors

1.How about the comparison between traditional and present work? It should address more motivation and real scenario. It could help to further know how to apply. 

2. The choices of wing type and cross section are not proper. The delta wing whith sweep is mainly adopted for supersonic airplane. Does it suitable for the sample of application?

3.There is lack of literation analysis.  And the number of grid is not also addressed. 

4.It should include more information about simulation in conclusions. It should look like a extented abstract. 

Comments on the Quality of English Language

There are many wording problems, such as "we" and passive expression.  

Reviewer 2 Report

Comments and Suggestions for Authors

The paper in question has the main critics are as follows:

1-      The abstract contains much technical information. It should be focuses on the main target of the research, brief on the methodology and the key results.   

2-      The introduction section should start with problem background of the current topic and state its importance.

3-      In order to keep the reader interested and give a comprehensive comprehension, strike a balance between technical insights and wider ramifications while describing the results.

4-      Make certain that every table and figure is understandable, appropriately labeled, and clearly related to the content. For improved comprehension, think about making complicated graphs simpler or using annotations.

5-      I think it is better to add a flowchart in order to summarize the methodology described.

6-      Ensure that all symbols and abbreviations are defined throughout the paper.

7-      Make suggestions for possible future work.

Comments on the Quality of English Language

-To make the document understandable to a wider audience, introduce technical terms and concepts with concise explanations.

-To improve readability, divide lengthy phrases into shorter, more concise statements.

Reviewer 3 Report

Comments and Suggestions for Authors

This paper proposes and validates a data-driven sparse sensor placement optimization algorithm for optimizing the position of flow sensors on UAV wings to achieve accurate flight state prediction and control, which has led to the development of bio-like "flight-aware" control systems. It is recommended to publish after some revisions:

1.       It is suggested to further clarify the limitations of existing sensor optimization methods in complex flow fields, highlight the specific problems and application scenarios solved in this study.

2.       Some latest literatures are suggested to be included into the Introduction to reflect the latest progress of data-driven optimization. See “, Machine Learning and Data-Driven Techniques for the Control of SmartPower Generation Systems: An Uncertainty Handling Perspective”.

3.       More quantitative indicators should be used to evaluate the performance improvement of the algorithm.

4.       "Incorporating a flowchart that illustrates the SSPOP algorithm would be highly beneficial.

5.       The discussion section can further emphasize the key performance indicators and analyze the causes of experimental phenomena.

6.       In the conclusion, it is suggested to add the prospect of the practical application of the algorithm and describe its application potential in complex aircraft.

Comments on the Quality of English Language

minor

Reviewer 4 Report

Comments and Suggestions for Authors

1. There are some issues with the writing of the Abstract:

- What are the issues with Sensor Placement and why it needs to be optimized? This is not clearly stated in the abstract;

- The abstract mentions the SSPOP algorithm, but its principle in solving the proposed problem is not clearly explained;

- What is the Bioinspired Approach and what is its principle? This is also not clearly explained.

- The Abstract lacks quantitative result analysis. Please use data to demonstrate the effectiveness of the proposed method.

- There are too many keywords in the Abstract, and it is recommended to reduce them to 5.

In summary, the Abstract of the article needs to be rewritten.

2. The authors mentions "greedy search algorithm and gradient based optimization". Please provide additional information on their current research status and existing problems in solving the proposed problems.

3. The Abstract mentions the comparison between the proposed method and the Greedy Search Algorithm and Gradient Based Optimization Algorithm. Please provide relevant experimental evidence to support this conclusion.

4. Rewrite the conclusion to highlight the results of the work done in this paper.

5. Polish the English writing.

Comments on the Quality of English Language

English language is required to be polished.

Round 2

Reviewer 2 Report

Comments and Suggestions for Authors

The authors have addressed all the comments.

Comments on the Quality of English Language

Minor editing of English language required.

Author Response

We have carefully reviewed the paper again and have made small edits as needed to enhance clarity.

Reviewer 4 Report

Comments and Suggestions for Authors

This paper can be accepted.

Comments on the Quality of English Language

Need to be polished further.

Author Response

(The authors gave the same response as above.)
